# Completely random measures for modelling block-structured sparse networks

**Tue Herlau**    **Mikkel N. Schmidt**    **Morten Mørup**
DTU Compute
Technical University of Denmark
Richard Petersens plads 31,
2800 Lyngby, Denmark
{tuhe,mns,mmor}@dtu.dk

## Abstract

Statistical methods for network data often parameterize the edge-probability by attributing latent traits such as block structure to the vertices and assume exchangeability in the sense of the Aldous-Hoover representation theorem. These assumptions are however incompatible with traits found in real-world networks such as a power-law degree-distribution. Recently, Caron & Fox (2014) proposed the use of a different notion of exchangeability after Kallenberg (2005) and obtained a network model which permits edge-inhomogeneity, such as a power-law degree-distribution whilst retaining desirable statistical properties. However, this model does not capture latent vertex traits such as block-structure. In this work we re-introduce the use of block-structure for network models obeying Kallenberg's notion of exchangeability and thereby obtain a collapsed model which both admits the inference of block-structure and edge inhomogeneity. We derive a simple expression for the likelihood and an efficient sampling method. The obtained model is not significantly more difficult to implement than existing approaches to block-modelling and performs well on real network datasets.

## 1 Introduction

Two phenomena are generally considered important for modelling complex networks. The first is community or block structure, where the vertices are partitioned into non-overlapping blocks (denoted by $\ell = 1, \ldots, K$ in the following) and the probability two vertices $i, j$ are connected depends on their assignment to blocks:

$$P\big(\text{Edge between vertex } i \text{ and } j\big) = \xi_{\ell m}$$

where $\xi_{\ell m} \in [0, 1]$ is a number only depending on the blocks $\ell, m$ to which $i, j$ respectively belongs. *Stochastic block models* (SBMs) were first proposed by White et al. (1976) and today form the basic starting point for many important link-prediction methods such as the *infinite relational model* (Xu et al., 2006; Kemp et al., 2006).

While block-structure is important for link prediction, the degree distribution of edges in complex networks is often found to follow a power-law (Newman et al., 2001; Strogatz, 2001). This realization has led to many important models of network *growth*, such as the preferential attachment (PA) model of Barabási (1999).

Models such as the IRM and the PA model have different goals. The PA model attempts to explain how network structure, such as the degree distribution, follows from simple rules of network growth and is not suitable for link prediction. In contrast, the IRM aims to discover latent block-structure

and predict edges — tasks for which the PA model is unsuitable. In the following, *network model* will refer to a model with the same aims as the IRM, most notably prediction of missing edges.

## 1.1 Exchangeability

Invariance is an important theme in Bayesian approaches to network modelling. For network data, the invariance which has received most attention is infinite exchangeability of random arrays. Suppose we represent the network as a subset of an infinite matrix $A = (A_{ij})_{ij \geq 1}$ such that $A_{ij}$ is the number of edges between vertex $i$ and $j$ (we will allow multi and self-edges in the following). Infinite exchangeability of the random array $(A_{ij})_{ij \geq 1}$ is the requirement that (Hoover, 1979; Aldous, 1981) $(A_{ij})_{ij \geq 1} \stackrel{d}{=} (A_{\sigma(i)\sigma(j)})_{ij \geq 1}$ for all finite permutations $\sigma$ of $\mathbb{N}$. The distribution of a finite network is then obtained by marginalization. According to the Aldous-Hoover theorem (Hoover, 1979; Aldous, 1981), an infinite exchangeable network has a representation in terms of a random function, and furthermore, the number of edges in the network must either scale as the square of the number of vertices or (with probability 1) be zero (Orbanz & Roy, 2015). Neither of these options are compatible with a power-law degree distribution and one is faced with the dilemma of giving up either the power-law distribution or exchangeability. It is the first horn of this dilemma which has been pursued by much work on Bayesian network modelling (Orbanz & Roy, 2015).

It is, however, possible to substitute the notation of infinite exchangeability in the above sense with a different definition due to Kallenberg (2005, chapter 9). The new notion retains many important characteristics of the former, including a powerful representation theorem parallelling the Aldous-Hoover theorem but expressed in terms of a random set. Important progress in exploring network models based on this representation has recently been made by Caron & Fox (2014), who demonstrate the ability to model power-law behaviour of the degree distribution and construct an efficient sampler for parameter inference. The reader is encouraged to consult this reference for more details.

In this paper, we will apply the ideas of Caron & Fox (2014) to block-structured network data, thereby obtaining a model based on the same structural invariance, yet able to capture both block-structure and degree heterogeneity. The contribution of this work is fourfold: (i) we propose general extension of sparse networks to allow latent structure, (ii) using this construction we implement a block-structured network model which obey Kallenbergs notion of exchangeability, (iii) we derive a collapsed expression of the posterior distribution which allows efficient sampling, (iv) demonstrate that the resulting model offers superior link prediction compared to both standard block-modelling and the model of Caron & Fox (2014).

It should be noted that independently of this manuscript, Veitch & Roy (2015) introduced a construction similar to our eq. (4) but focusing on the statistical properties of this type of random process, whereas this manuscript focuses on the practical implementation of network models based on the construction.

## 2 Methods

Before introducing the full method we will describe the construction informally, omitting details relating to completely random measures.

### 2.1 A simple approach to sparse networks

Suppose the vertices in the network are labelled by real numbers in $\mathbb{R}_+$. An edge $e$ (edges are considered directed and we allow for self-edges) then consists of two numbers $(x_{e1}, x_{e2}) \in \mathbb{R}_+^2$ denoted the *edge endpoint*. A network $X$ of $L$ edges (possibly $L = \infty$) is simply the collection of points $X = ((x_{e1}, x_{e2}))_{e=1}^L \subset \mathbb{R}_+^2$. We adopt the convention that multi-edges implies duplicates in the list of edges. Suppose $X$ is generated by a Poisson process with base measure $\xi$ on $\mathbb{R}_+^2$

$$X \sim \mathrm{PP}(\xi). \tag{1}$$

A *finite* network $X_\alpha$ can then be obtained by considering the restriction of $X$ to $[0, \alpha]^2$: $X_\alpha = X \cap [0, \alpha]^2$. As an illustration, suppose $\xi$ is the Lebesgue measure. The number of edges is then $L \sim \mathrm{Poisson}(\alpha^2)$ and the edge-endpoints $x_{e1}, x_{e2}$ are i.i.d. on $[0, \alpha]$ simply corresponding to selecting $L$ random points in $[0, \alpha]^2$. The edges are indicated by the gray squares in figure 1a and the

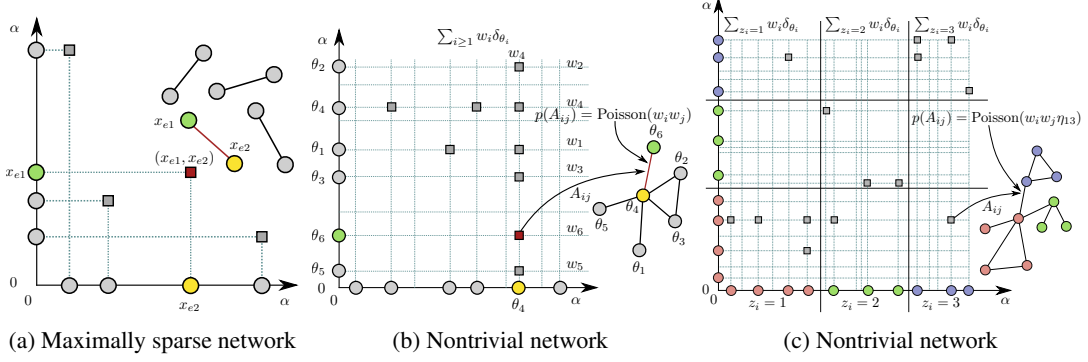

(a) Maximally sparse network     (b) Nontrivial network     (c) Nontrivial network

Figure 1: (Left:) A network is generated by randomly selecting points from $[0, \alpha]^2 \subset \mathbb{R}_+^2$ corresponding to edges (squares) and identifying the unique coordinates with vertices (circles), giving the maximally disconnected graph. (Middle:) The edges are restricted to lie at the intersection of randomly generated gray lines at $\theta_i$, each with a mass/sociability parameter $w_i$. The probability of selecting an intersection is proportional to $w_i w_j$, giving a non-trivial network structure. (Right:) Each vertex is assigned a latent trait $z_i$ (the assignment to blocks as indicated by the colors) that modulates the edge probability with a parameter $\eta_{\ell m} \geq 0$, thus allowing block-structured networks.

vertices as circles. Notice the vertices will be distinct with probability 1 and the procedure therefore gives rise to the degenerate but *sparse* network of $2L$ vertices and $L$ edges, shown in figure 1a.

To generate non-trivial networks, the edge-endpoints must coincide with nonzero probability. Similar to Caron & Fox (2014), suppose the coordinates are restricted to only take a countable number of potential values, $\theta_1, \theta_2, \cdots \in \mathbb{R}_+$ and each value has an associated *sociability* (or *mass*) parameter $w_1, w_2, \cdots \in [0, \infty[$ (we use the shorthand $(\theta_i)_i = (\theta_i)_{i=1}^\infty$ for a series). If we define the measure $\mu = \sum_{i \geq 1} w_i \delta_{\theta_i}$ and let $\xi = \mu \times \mu$, then generating $X_\alpha$ according to the procedure of eqn. (1) the number of edges $L$ is $\mathrm{Poisson}(T^2)$, $T = \mu([0, \alpha]) = \sum_{i=1}^\infty w_i$ distributed. The position of the edges remains identically distributed, but with probability proportional to $w_i w_j$ of selecting coordinate $(\theta_i, \theta_j)$. Since the edge-endpoints coincide with non-zero probability this procedure allows the generation of a non-trivial associative network structure, see figure 1b. With proper choice of $(w_i, \theta_i)_{i \geq 1}$ these networks exhibit many desirable properties, such as a power-law degree distribution and sparsity (Caron & Fox, 2014).

This process can be intuitively extended to block-structured networks, as illustrated in figure 1c. There, each vertex is assigned a *latent trait* (i.e. a block assignment), here highlighted by the colors. We use the symbol $z_i \in \{1, \ldots, K\}$ to indicate the assignment of vertex $i$ to one of the $K$ blocks. We can then consider a measure of the form

$$\xi = \sum_{i,j \geq 1} \eta_{z_i z_j} w_i w_j \delta_{(\theta_i, \theta_j)} = \sum_{\ell, m = 1}^K \eta_{\ell m} \mu_\ell \times \mu_m, \tag{2}$$

where we have introduced $\mu_\ell = \sum_{i : z_i = \ell} w_i \delta_{\theta_i}$. Defined in this manner, $\xi$ is a measure on $[0, \alpha]^2$ and $\eta_{\ell m}$ parameterizes the interaction strength between community $\ell$ and $m$. Notice the number of edges $L_{\ell m}$ between block $\ell$ and $m$ is, by basic properties of the Poisson process, distributed as $L_{\ell m} \sim \mathrm{Poisson}(\eta_{\ell m} T_\ell T_m)$, where $T_\ell = \mu_\ell([0, \alpha])$. In figure 1c the locations $\theta_i$ of the vertices have been artificially ordered according to color for easy visualization. The following section will show the connection between the above construction of eq. (2) and the exchangeable representation due to Kallenberg (2005). However, for greater generality, we will let the latent trait be a general continuous parameter $u_i \in [0, 1]$ and later show that block-structured models can be obtained as a special case.

## 2.2 Exchangeability and point-process network models

Since the networks in the point-set representation are determined by the properties of the measure $\xi$, invariance (i.e. exchangeability) of random point-set networks is defined as invariance of this random measure. Recall infinite exchangeability for infinite matrices requires that the distribution of the random matrix to be unchanged by permutation of the rows/columns in the network. For

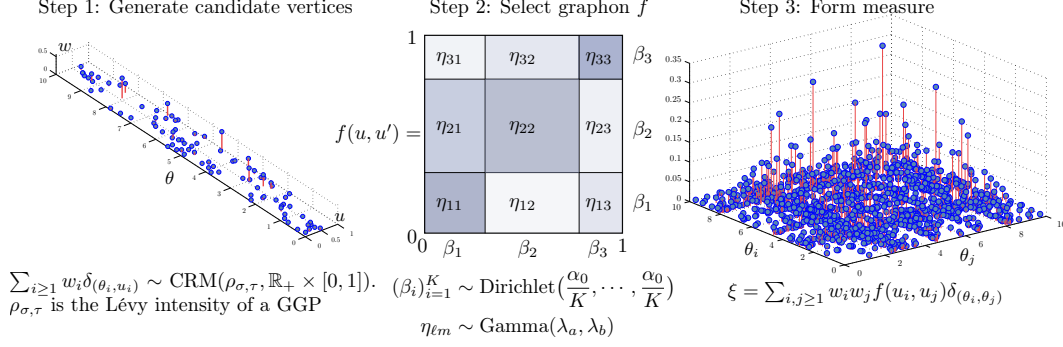

Step 1: Generate candidate vertices     Step 2: Select graphon $f$     Step 3: Form measure

$\sum_{i \geq 1} w_i \delta_{(\theta_i, u_i)} \sim \mathrm{CRM}(\rho_{\sigma,\tau}, \mathbb{R}_+ \times [0,1])$.
$\rho_{\sigma,\tau}$ is the Lévy intensity of a GGP

$(\beta_i)_{i=1}^K \sim \mathrm{Dirichlet}\left(\frac{\alpha_0}{K}, \cdots, \frac{\alpha_0}{K}\right)$

$\xi = \sum_{i,j \geq 1} w_i w_j f(u_i, u_j) \delta_{(\theta_i, \theta_j)}$

$\eta_{\ell m} \sim \mathrm{Gamma}(\lambda_a, \lambda_b)$

Figure 2: (Step 1:) The potential vertex locations, $\theta_i$, latent traits $u_i$ and sociability parameters $w_i$ are generated using a *generalized gamma process* (Step 2:) The interaction of the latent traits $f : [0,1]^2 \to \mathbb{R}_+$, *the graphon*, is chosen to be a piece-wise constant function (Step 3:) Together, these determine the random measure $\xi$ which is used to generate the network from a Poisson process

a random measure on $\mathbb{R}_+^2$, the corresponding requirement is that it should be possible to partition $\mathbb{R}_+$ into intervals $I_1, I_2, I_3, \ldots$, permute the intervals, and have the random measure be invariant to this permutation. Formally, a random measure $\xi$ on $\mathbb{R}_+^2$ is then said to be *jointly exchangeable* if $\xi \circ (\varphi \otimes \varphi)^{-1} \overset{d}{=} \xi$ for all measure-preserving transformations $\varphi$ of $\mathbb{R}_+$. According to Kallenberg (2005, theorem 9.24), this is ensured provided the measure has a representation of the form:

$$\xi = \sum_{i,j \geq 1} h(\zeta, x_i, x_j) \delta_{(\theta_i, \theta_j)}, \tag{3}$$

where $h$ is a measurable function, $\zeta$ is a random variable and $\{(x_i, \theta_i)\}_{i \geq 1}$ is a unit rate Poisson process on $\mathbb{R}_+^2$ (the converse involves five additional terms (Kallenberg, 2005)). In this representation, the locations $(\theta_i)_i$ and the parameters $(x_i)_i$ are decoupled, however we are free to select the random parameters $(x_i)_{i \geq 1}$ to lie in a more general space than $\mathbb{R}_+$. Specifically, we define

$$x_i = (u_i, v_i) \in [0,1] \times \mathbb{R}_+,$$

with the interpretation that each $v_i$ corresponds to a random mass $w_i$ through a transformation $w_i = g(v_i)$, and each $u_i \in [0,1]$ is a general *latent trait* of the vertex. (In figure 1 this parameter corresponded to the assignment to blocks). We then consider the following choice:

$$h(\zeta, x_i, x_j) = f(u_i, u_j) g_{z_i}(v_i) g_{z_j}(v_j) \tag{4}$$

where $f : [0,1]^2 \to \mathbb{R}_+$ is a measurable function playing a similar role as the graphon in the Aldous-Hoover representation, and $\{(u_i, v_i, \theta_i)\}_{i \geq 1}$ follows a unit-rate Poisson process on $[0,1] \times \mathbb{R}_+^2$.

To see the connection with the block-structured model, suppose the function $f$ is a piece-wise constant function

$$f(u, u') = \sum_{\ell, m = 1}^K \eta_{\ell m} 1_{J_\ell}(u) 1_{J_m}(u'),$$

where $J_\ell = \left[ \sum_{m=1}^{\ell-1} \beta_m, \sum_{m=1}^{\ell} \beta_m \right[$, $\sum_{\ell=1}^K \beta_\ell = 1$, $\beta_\ell > 0$ and $z_i = \ell$ denotes the event $1_{J_\ell}(u_i) = 1$. Notice this choice for $f$ is exactly equivalent to the graphon for the block-structured network model in the Aldous-Hoover representation (Orbanz & Roy, 2015). The procedure is illustrated in figure 2. Realizations of networks generated by this process using different values of $K$ can be obtained using the simulation methods of Caron & Fox (2014) and can be seen in figure 3. Notice the $K = 1, \eta_{11} = 1$ case corresponds to their method.

To fully define the method we must first introduce the relevant prior for the measure $\mu = \sum_{i \geq 1} w_i \delta_{(\theta_i, u_i)}$. As a prior we will use the Generalized Gamma-process (GGP) (Hougaard, 1986). In the following section, we will briefly review properties of completely random measures and use these to derive a simple expression of the posterior.

## 2.3 Random measures

As a prior for $\mu$ we will use completely random measures (CRMs) and the reader is referred to (Kallenberg, 2005; Kingman, 1967) for a comprehensive account. Recall first the definition of a CRM. Assume $\mathbb{S}$ is a separable complete metric space with the Borel $\sigma$-field $\mathcal{B}(\mathbb{S})$ (for our purpose $\mathbb{S} = [0, \alpha]$). A *random measure* $\mu$ is a random variable whose values are measures on $\mathbb{S}$. For each measurable set $A \in \mathcal{B}(\mathbb{S})$, the random measure induces a random variable $\mu(A)$, and the random measure $\mu$ will be said to be *completely random* if for any finite collection $A_1, \ldots, A_n$ of disjoint measurable sets the random variables $\mu(A_1), \ldots, \mu(A_n)$ are independent. It was shown by Kingman (1967) that the non-trivial part of any random measure $\mu$ is discrete almost certainly with a representation

$$\mu = \sum_{i=1}^{\infty} w_i \delta_{\theta_i}, \qquad (5)$$

where the sequence of *masses* and *locations* $(w_i, \theta_i)_i$ (also known as the *atoms*) is a Poisson random measure on $\mathbb{R}^+ \times \mathbb{S}$, with mean measure $\nu$ known as the Lévy intensity measure. We will consider *homogeneous* CRMs, where locations are independent, $\nu(dw, d\theta) = \rho(dw)\kappa_\alpha(d\theta)$, and assume $\kappa_\alpha$ is the Lebesgue measure on $[0, \alpha]$.

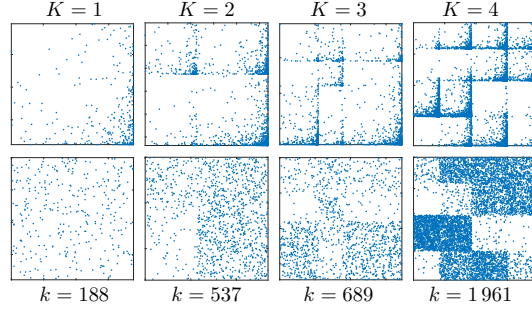

Figure 3: (Top:) Example of four randomly generated networks for $K = 1, 2, 3$ and $4$ using the choice of random measure discussed in section 2.3. The other parameters were fixed at $\alpha = 20K, \tau = 1, \sigma = 0.5$ and $\lambda_a = \lambda_b = 1$. Vertices have been sorted according to their assignment to blocks and sociability parameters.(Bottom:) The same networks as above but applying a random permutation to the edges within each tile. A standard SBM assumes a network structure of this form.

Since the construction as outlined in figure 1c depends on sampling the edge start and end-points at random from the locations $(\theta_i)_i$, with probability proportional to $w_i$, the normalized form of eqn. (5) will be of particular interest. Specifically, the chance of selecting a particular location from a random draw is governed by

$$P = \frac{\mu}{T} = \sum_{i=1}^{\infty} p_i \delta_{\theta_i}, \quad p_i = \frac{w_i}{T}, \quad T = \mu(\mathbb{S}) = \sum_{i=1}^{\infty} w_i, \qquad (6)$$

which is known as the *normalized random measure* (NRM) and $T$ is the *total mass* of the CRM $\mu$ (Kingman, 1967). A random draw from a Poisson process based on the CRM can thus be realized by first sampling the number of generated points, $L \sim \text{Poisson}(T)$, and then drawing their locations in a i.i.d. manner from the NRM of eqn. (6). The reader is referred to James (2002) for a comprehensive treatment on NRMs.

With the notation in place, we can provide the final form of the generative process for a network $X_\alpha$. Suppose the CRM $\mu$ (restricted to the region $[0, \alpha]$) has been generated. Assume $z_i = \ell$ iff. $u_i \in J_\ell$ and define the $K$ thinned measures on $[0, \alpha]$ as:

$$\mu_\ell = \sum_{i:z_i=\ell} w_i \delta_{\theta_i}$$

each with total mass $T_\ell = \mu_\ell([0, \alpha])$. By basic properties of CRMs, the thinned measures are also CRMs (Pitman, 2006). The number of points in each tile $L_{\ell m}$ is then $\text{Poisson}(\eta_{\ell m} T_\ell T_m)$ distributed, and given $L_{\ell m}$ the edge-endpoints $(x_{e1\ell}, x_{e2m})$ between atoms in measure $\ell$ and $m$ can then be drawn from the corresponding NRM. The generative process is then simply:

$$(\beta_\ell)_{\ell=1}^{K} \sim \text{Dirichlet}(\beta_0/K, \ldots, \beta_0/K) \qquad \mu \overset{\text{iid}}{\sim} \text{CRM}(\rho, U_{[0,1]} \times U_{\mathbb{R}_+})$$

$$\eta_{\ell k} \overset{\text{iid}}{\sim} \text{Gamma}(\lambda_a, \lambda_b) \qquad L_{\ell m} \overset{\text{iid}}{\sim} \text{Poisson}(\eta_{\ell m} T_\ell T_m)$$

$$\text{for } e = 1, \ldots, L_{\ell m}: x_{e1\ell} \overset{\text{iid}}{\sim} \text{Categorical}((w_i/T_\ell)_{z_i=\ell}) \qquad x_{e2m} \overset{\text{iid}}{\sim} \text{Categorical}((w_j/T_m)_{z_j=m}).$$

In the following we will use the *generalized gamma process* (GGP) as the choice of Lévy intensity measure (James, 2002). The GGP is parameterized with two parameters $\sigma, \tau$ and has the functional form

$$\rho_{\sigma,\tau}(dw) = \frac{1}{\Gamma(1-\sigma)} w^{-1-\sigma} e^{-\tau w} dw.$$

The parameters $(\sigma, \tau)$ will be restricted to lie in the region $]0, 1[\times[0, \infty[$ as in (Caron & Fox, 2014). In conjunction with $\alpha$ we thus obtain three parameters $(\alpha, \sigma, \tau)$ which fully describe the CRM and the induced partition structure.

## 2.4 Posterior distribution

In order to define a sampling procedure of the CRMSBM we must first characterize the posterior distribution. In Caron & Fox (2014) this was calculated using a specially tailored version of Palm's formula. In this work we will use a counting argument inspired by Pitman (2003, eqn. (32)) and a reparameterization to collapse the weight-parameter $(w_i)_{i\geq 1}$ to obtain a fairly simple analytical expression which is amenable to standard sampling procedures. The full derivation is, however, somewhat lengthy and is included in the supplementary material.

First notice the distribution of the total mass $T_\ell$ of each of the thinned random measures $\mu_\ell$ is a tilted $\sigma$-stable random variable (Pitman, 2006). If we introduce $\alpha_\ell \equiv \beta_\ell \alpha$, its density $g_{\alpha_\ell,\sigma,\tau}$ may be written as

$$g_{\alpha,\sigma,\tau}(t) = \theta^{-\frac{1}{\sigma}} f_\sigma(t\theta^{-\frac{1}{\sigma}}) \phi_\lambda(t\theta^{-\frac{1}{\sigma}})$$

where $\phi_\lambda(t) = e^{\lambda^\sigma - \lambda t}$, $\lambda = \tau\theta^{\frac{1}{\sigma}}$, $\theta = \frac{\alpha}{\sigma}$ and $f_\sigma$ is the density of a $\sigma$-stable random variable. See Devroye & James (2014) for more details. According to Zolotarev's integral representation, the function $f_\sigma$ has the following form (Zolotarev, 1964)

$$f_\sigma(x) = \frac{\sigma x^{\frac{-1}{1-\sigma}}}{\pi(1-\sigma)} \int_0^\pi du\, A(\sigma, u) e^{\frac{-A(\sigma,u)}{x^{\sigma/(1-\sigma)}}}, \quad A(\sigma, u) = \sin((1-\sigma)u) \left[\frac{\sin(\sigma u)^\sigma}{\sin(u)}\right]^{\frac{1}{1-\sigma}}. \quad (7)$$

Since not all potential vertices (i.e. terms $w_i \delta_{\theta_i}$ in $\mu$) will have edges attached to them, it is useful to introduce a variable which encapsulates this distinction. We therefore define the variable $\tilde{z}_i = 0, 1, \ldots, K$ with the definition:

$$\tilde{z}_i = \begin{cases} z_i & \text{if there exists } (x, y) \in X_\alpha \text{ s.t. } \theta_i \in \{x, y\}, \\ 0 & \text{otherwise.} \end{cases}$$

In addition, suppose for each measure $\mu_\ell$, the end-points of the edges associated with this measure selects $k_\ell = |\{i : \tilde{z}_i = \ell\}|$ unique atoms and $k = \sum_{\ell=1}^K k_\ell$ is the total number of vertices in the network. Next, we consider a specific network $(A_{ij})_{i,j=1}^k$ and assume it is labelled such that atom $(w_i, \theta_i)$ corresponds to a particular vertex $i$ in the network. We also define $n_i = \sum_j (A_{ij} + A_{ji})$ as the number of edge-endpoints that selects atom $i$, $n_\ell = \sum_{i:\tilde{z}_i=\ell} n_i$ as the aggregated edge-endpoints that select measure $\mu_\ell$ and $n_{\ell m} = \sum_{\tilde{z}_i=\ell, z_m=j} A_{ij}$ as the edges between measure $\mu_\ell$ and $\mu_m$. The posterior distribution is then

$$P(A, (z_i)_i, \sigma, \tau, (\alpha_\ell, s_\ell, t_\ell)_\ell) = \frac{\Gamma(\beta_0) \prod_{\ell=1}^K \alpha_\ell^{\frac{\beta_0}{K}-1} E_\ell}{\Gamma(\frac{\beta_0}{K})^K \alpha^{\beta_0} \prod_{ij} A_{ij}!} \prod_{\ell m} \frac{G(\lambda_a + n_{\ell m}, \lambda_b + T_\ell T_m)}{G(\lambda_a, \lambda_b)}, \quad (8)$$

where we have introduced:

$$E_\ell = \frac{\alpha^{k_\ell} s_\ell^{n_\ell - k_\ell \sigma - 1}}{\Gamma(n_\ell - k_\ell \sigma) e^{\tau s_\ell}} g_{\alpha_\ell, \tau, \sigma}(T_\ell - s_\ell) \prod_{\tilde{z}_i=\ell} (1-\sigma)_{n_i}$$

and $s_\ell = \sum_{i:\tilde{z}_i=\ell} w_i$ is the mass of the "occupied" atoms in the measure $\mu_\ell$. The posterior distribution can be seen as the product of $K$ partition functions corresponding to the GGP, multiplied by the $K^2$ interaction factors involving the function $G(a, b) = \Gamma(a) b^{-a}$, and corresponding to the interaction between the measures according to the block structure assumption.

Note that the $\eta = 1$ case, corresponding to a collapsed version of Caron & Fox (2014), can be obtained by taking the limit $\lambda_a = \lambda_b \to \infty$, in which case $\frac{G(\lambda_a+n, \lambda_b+T)}{G(\lambda_a, \lambda_b)} \to e^{-T}$. When discussing the $K = 1$ case, we will assume this limit has been taken.

## 2.5 Inference

Sampling the expression eqn. (8) requires three types of sampling updates: (i) the sequence of block-assignments $(z_i)_i$ must be updated, (ii) in the simulations we will consider binary networks and we will therefore need to both impute the integer valued counts (if $A_{ij} > 0$), as well as missing values in the network, and (iii) both the parameters associated with the random measure, $\sigma$ and $\tau$, as well as the remaining variables associated with each expression $E_\ell$ must be updated.

All terms, except the densities $g_{\alpha,\sigma,\tau}$, are amenable to standard sampling techniques. We opted for the approach of Lomelí et al. (2014), in which $u$ in Zolotarev's integral representation (eqn. 7) is considered an auxiliary parameter. The full inference procedure can be found in the supplementary material, however, the main steps are: [1]

**Update of $(z_i)_i$:** For each $\ell$, impute $(w_i)_{\bar{z}_i = \ell}$ once per sweep (see supplementary for details), and then iterate over $i$ and update each $z_i$ using a Gibbs sweep from the likelihood. The Gibbs sweep is no more costly than that of a standard SBM.

**Update of $A$:** Impute $(\eta_{\ell m})_{\ell m}$ and $(w_i)_i$ once per sweep (see supplementary for details), and then for each $(ij)$ such that the edge is either unobserved or must be imputed ($A_{ij} \geq 1$), generate a candidate $a \sim \text{Poisson}(\eta_{\ell m} w_i w_j)$. Then, if the edge is unobserved, simply set $A_{ij} = a$, otherwise if the edge is observed and $a = 0$, reject the update.

**Update of $\sigma, \tau$:** For $\ell = 1, \ldots, K$, introduce $u_\ell$ corresponding to $u$ in Zolotarev's integral representation (eqn. 7) and let $t_\ell = T_\ell - s_\ell$. Update the four variables in $\Phi_\ell = (\alpha_\ell, u_\ell, s_\ell, t_\ell)$ and $\sigma, \tau$ using random-walk Metropolis Hastings updates.

In terms of computational cost, the inference procedure is of the same order as the SBM albeit with higher constants due to the overall complexity of the likelihood and because the parameters $(\alpha_\ell, u_\ell, s_\ell, t_\ell)$ must be sampled for each CRM. In Caron & Fox (2014), the parameters $(w_i)_{i \geq 1}$ were sampled using Hamiltonian Monte Carlo, whereas herein they are collapsed and re-imputed.

The parameters $\Phi_\ell$ and $\sigma, \tau$ are important for determining the sparsity and power-law properties of the network model (Caron & Fox, 2014). To investigate convergence of the sampler for these parameters, we generated a single network problem using $\alpha = 25, \sigma = 0.5, \tau = 2$ and evaluated 12 samplers with $K = 1$ on the problem. Autocorrelation plots (mean and standard deviation computed over 12 restarts) can be seen in figure 4a. All parameters mix, however the different parameters have different mixing times with $u$ in particular being affected by excursions. This indicates many sampling updates of $\Phi_\ell$ are required to explore the state space sufficiently and we therefore applied 50 updates of $\Phi_\ell$ for each update of $(z_i)_i$ and $A_{ij}$. Additional validation of the sampling procedure can be found in the supplementary material.

## 3 Experiments

The proposed method was evaluated on 11 network datasets (a description of how the datasets were obtained and prepared can be found in the supplementary material) using $K = 200$ in the truncated stick-breaking representation. As a criteria of evaluation we choose AUC score on held-out edges, i.e. predicting the presence or absence of unobserved edges using the imputation method described in the previous section. All networks were initially processed by thresholds at 0, and vertices with zero edges were removed. A fraction of 5% of the edges were removed and considered as held-out data.

To examine the effect of using blocks, we compared the method against the method of Caron & Fox (2014) (CRM) (corresponding to $\eta_{\ell m} = 1$ and $K = 1$), a standard block-structured model with Poisson observations (pIRM) (Kemp et al., 2006), and the degree-corrected stochastic block model (DCSBM) Herlau et al. (2014). The later allows both block-structure and degree-heterogeneity but it is not exchangeable. More details on the simulations and methods are found in the supplementary material.

The pIRM was selected since it is the closest block-structured model to the CRMSBM without degree-correction. This allows us to determine the relative benefit of inferring the degree-distribution compared to only the block-structure. For the priors we selected uniform priors for $\sigma, \tau, \alpha$ and a $\text{Gamma}(2, 1)$ prior for $\beta_0, \lambda_a, \lambda_b$. Similar choices were made for the other models.

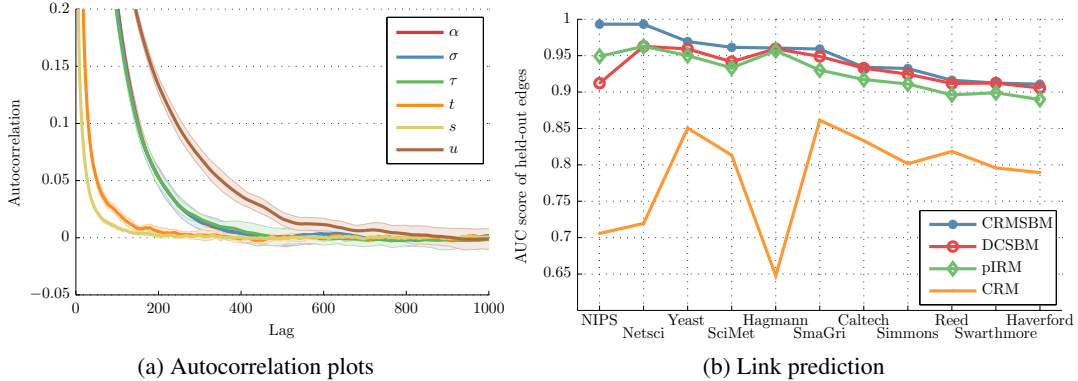

| (a) Autocorrelation plots | (b) Link prediction |

Figure 4: (Left:) Autocorrelation plots of the parameters $\alpha, \sigma, \tau, s, t$ and $u$ for a $K = 1$ network drawn from the prior distribution using $\alpha = 25$, $\sigma = 0.5$ and $\tau = 2$. The plots were obtained by evaluating the proposed sampling procedure for $10^6$ iterations and the shaded region indicates standard deviation obtained over 12 re-runs. The simulation indicates reasonable mixing for all parameters, with $u$ being the most affected by excursions. (Right:) AUC score on held-out edges for the selected methods (averaged over 4 restarts) on 11 network datasets. For the same number of blocks, the CRMSBM offers good link-prediction performance compared to the method of Caron & Fox (2014) (CRM), a SBM with Poisson observations (pIRM) and the degree-corrected SBM (DCSBM) (Herlau et al., 2014). Additional information is found in the supplementary material.

All methods were evaluated for $T = 2\,000$ iterations, and the latter half of the chains was used for link prediction. We used 4 random selections of held-out edges per network to obtain the results seen in figure 4b (same sets of held-out edges were used for all methods). It is evident that block-structure is crucial to obtain good link prediction performance. For the block-structured methods, the results indicate additional benefits from using models which permits degree-heterogenity upon most networks, except the Hagmann brain connectivity graph. This result is possibly explained by the Hagmann graph having little edge-inhomogeneity. Comparing the CRMSBM and the DCSBM, these models perform either on par with or with a slight advantage to the CRMSBM.

## 4 Discussion and Conclusion

Models of networks based on the CRM representation of Kallenberg (2005) offer one of the most important new ideas in statistical modelling of networks in recent years. To our knowledge Caron and Fox (2014) were the first to realize the benefits of this modelling approach, describe its statistical properties and provide an efficient sampling procedure.

The degree distribution of a network is only one of several important characteristics of a complex network. In this work we have examined how the ideas presented in Caron and Fox (2014) can be applied for a simple block-structured network model to obtain a model which admits block structure and degree correction. Our approach is a fairly straightforward generalization of the methods of Caron and Fox (2014). However, we have opted to explicitly represent the density of the total mass $g_{\alpha_\ell, \sigma, \tau}$ and integrate out the sociability parameters $(w_i)_i$, thereby reducing the number of parameters associated with the CRM from the order of vertices to the order of blocks.

The resulting model has the increased flexibility of being able to control the degree distribution within each block. In practice, results of the model on 11 real-world datasets indicate that this flexibility offers benefits over purely block-structured approaches to link prediction for most networks, as well as potential benefits over alternative approaches to modelling block-structure and degree-heterogeneity. The results strongly indicate that structural assumptions (such as block-structure) are important to obtain reasonable link prediction.

Block-structured network modelling is in turn the simplest structural assumption for block-modelling. The extension of the method of Caron and Fox (2014) to overlapping blocks, possibly using the dependent random measures of Chen et al. (2013), appears fairly straightforward and should potentially offer a generalization of overlapping block models.

**Acknowledgments**

This project was funded by the Lundbeck Foundation (grant nr. R105-9813).

## Footnotes

[1]Code available at `http://people.compute.dtu.dk/tuhe/crmsbm`.

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
