[Supplementary Material · supplementary.pdf]

# Completely random measures for modelling block-structured sparse networks — supplementary material

**Tue Herlau**    **Mikkel N. Schmidt**    **Morten Mørup**
DTU Compute
Technical University of Denmark
Richard Petersens plads 31,
2800 Lyngby, Denmark
{tuhe,mmor,mnsc}@dtu.dk

**Derivation of posterior Distribution**

The posterior distribution will be derived using a counting argument inspired by Pitman (2003, eqn. (32)). Consider first the case where the interaction strengths $(\eta_{\ell m})_{\ell m}$ and block sizes $(\beta_\ell)_\ell$ has a fixed value and that number of edges $L_{\ell m}$ within each tile $(\ell, m)$ is given and recall the NRM is defined as:

$$P = \frac{\mu}{T} = \sum_{i=1}^{\infty} p_i \delta_{\theta_i}, \quad p_i = \frac{w_i}{T}, \quad T = \mu(\mathbb{S}) = \sum_{i=1}^{\infty} w_i. \tag{1}$$

Since not all potential vertices (i.e. terms $w_i \delta_{\theta_i}$ in $\mu$) will have edges attached to them it is useful to introduce a variable which encapsulates this distinction. We therefore define the variable $\tilde{z}_i = 0, 1, \ldots, K$ with the definition:

$$\tilde{z}_i = \begin{cases} z_i & \text{if there exist } (x,y) \in X_\alpha \text{ st. } \theta_i \in \{x,y\}, \\ 0 & \text{otherwise.} \end{cases}$$

Suppose in addition for each measure $\mu_\ell$, the end-points of the edges associated with this measure selects $k_\ell = |\{i : \tilde{z}_i = \ell\}|$ unique atoms and that the number of edge-endpoints selecting any particular atom $w_i$ is $n_i$. This naturally divides the edge-endpoints associated with a particular measure $\ell$ into a partition, $\{B_1, \ldots, B_{k_\ell}\}$ (Pitman, 2003), and we denote by $\Pi_{\ell, 2L}$ this random partition for measure $\ell$. For a particular measure the joint distribution

$$P(\Pi_{\ell,2L} = \{B_1, \ldots, B_{k_\ell}\}, w_i \in dw_i, T_\ell \in dT_\ell) \tag{2}$$

is obtained from three contributions (with $\alpha_\ell \equiv \beta_\ell \alpha$):

- The mass parameter $T_{\alpha_\ell}$ is distributed as $g_{\alpha_\ell, \sigma, \tau}$
- For each $\ell = 1, \ldots, K$, there must be a Poisson atom in $dw_i$ for each $i$ such that $\tilde{z}_i = \ell$
- For each $\ell$, we know there are Poisson atoms in $(dw_i)_{\tilde{z}_i = \ell}$, however since the measure of these intervals is infinitesimal, the remaining mass $T_\ell - \sum_{\tilde{z}_i = \ell} w_i$ is still distributed as $g_{\alpha_\ell, \sigma, \tau}$.
- Each edge-endpoint selects the atom independently with probability given by the NRM of eqn. (1), $w_i / T_\ell$.

The probability eqn. (2) can then be obtained from these three contributions as (with $k = \sum_{\ell=1}^K k_\ell$ being the total number of vertices in the network):

$$\left\{ \prod_{i=1}^{k} \alpha \rho_{\sigma,\tau}(dw_i) \right\} \prod_{\ell}^{K} \left\{ g_{\alpha_\ell,\sigma,\tau}(T_\ell - \sum_{i:\tilde{z}_i=\ell} w_i) \right\} \left\{ \prod_{i:\tilde{z}_i=\ell} \left( \frac{w_i}{T_\ell} \right)^{n_i} \right\} \tag{3}$$

where $n_i$ is the total number of times a particular atom $i$ of $\mu_\ell$ is selected in the process. To connect these definitions to actual network data, i.e. an array $(A_{ij})_{i,j=1}^k$, notice if the atom $(w_i, \theta_i)$ corresponds to a particular vertex $i$ in the network then $n_i = \sum_j (A_{ij} + A_{ji})$.

Returning to eqn. (3) for a particular $\ell$, the expression can be integrated by introducing the variables $s_\ell = \sum_{i:\tilde{z}_i=\ell} w_i$ corresponding to the sum of the *selected* atoms, introducing the parameters $x_i = w_i/s_{z_i}$, and integrating (Pitman, 2003; Lijoi et al., 2008; Favaro & Teh, 2013). With $n_\ell = \sum_{i:\tilde{z}_i=\ell} n_i$ eqn. (3) can be written as a product over $K$ factors:

$$\prod_{i:\tilde{z}_i=\ell}^{k} (1-\sigma)_{n_i} \int_0^{T_\ell} ds_\ell \frac{s_\ell^{n_\ell-k_\ell\sigma-1} g_{\alpha,\sigma,\tau}(T_\ell-s_\ell)}{\Gamma(n_\ell-k_\ell\sigma)T_\ell^{n_\ell}\alpha_\ell^{-k_\ell}e^{\tau s_\ell}}. \tag{4}$$

Recall the number of edges within each tile $L_{\ell m}$ is Poisson with rate $\eta_{\ell m}T_\ell T_m$. In addition, when considering a concrete observed data matrix the edges does not have a particular labelling which is otherwise introduced in the proceeding counting argument. Thus, if we observe a number $A_{ij}$ of edges between vertices $i, j$ in a particular tile, we must consider all ways a network with this number of edges can be obtained by our generative process. This is equivalent to the number of ways of selecting the particular edge-counts of the total edge-counts within each tile. The multiplicity becomes the multinomial coefficient:

$$\binom{L_{\ell m}}{(A_{ij})_{\tilde{z}_i=\ell,\tilde{z}_j=m}} = \frac{L_{\ell m}!}{\prod_{\tilde{z}_i=\ell,\tilde{z}_j=m} A_{ij}!}. \tag{5}$$

The probability of obtaining a particular observed network $A_{ij}$ can be obtained by combining eqs. (4), (5) and the Poisson rates for the edge-counts within each tile to obtain:

$$P(A, (z_i)_i | (\eta_{\ell m})_{\ell m}, (\beta_\ell)_\ell) = \left\{ \prod_{\ell=1}^{K} \int_0^\infty dT_\ell \{\text{eqn. (4)}\} \right\} \prod_{\ell m} \left\{ \text{Poisson}(L_\ell | \eta_{\ell m} T_\ell T_m) \frac{L_{\ell m}!}{\prod_{\substack{\tilde{z}_i=\ell, \\ \tilde{z}_j=m}} A_{ij}!} \right\}.$$

Defining $n_{\ell m} = \sum_{\tilde{z}_i=\ell,\tilde{z}_m=j} A_{ij}$ and simplifying

$$P(A, (z_i)_i | (\eta_{\ell m}), (\beta_\ell)_\ell) = \frac{1}{\prod_{ij} A_{ij}!} \prod_\ell \left[ \int_0^\infty \int_0^{T_\ell} dT_\ell ds_\ell \right] \left[ \prod_{\ell m} \eta_{\ell m}^{n_{\ell m}} e^{-\sum_{\ell m} \eta_{\ell m} T_\ell T_m} \right] \left\{ \prod_\ell E_\ell \right\}$$

where we have defined

$$E_\ell = \frac{\alpha^{k_\ell} s_\ell^{n_\ell-k_\ell\sigma-1}}{\Gamma(n_\ell-k_\ell\sigma)e^{\tau s_\ell}} g_{\alpha_\ell,\tau,\sigma}(T_\ell-s_\ell) \prod_{\tilde{z}_i=\ell} (1-\sigma)_{n_i}.$$

Similar to Lijoi et al. (2008) we will use the simple change-of-variable from $T$ to $t = T - s$ and a change in the order of integration to obtain:

$$\int_{\mathbb{R}^+} \int_0^s dT ds\, h(T, s) = \iint_{\mathbb{R}_+^2} ds dt\, h(t+s, s). \tag{6}$$

Then introducing the Gamma-priors for $\eta_{\ell m}$, Dirichlet prior for $(\beta_\ell)_\ell$ and integrating over $\eta_{\ell m}$ we obtain the final expression:

$$P(A, (z_i)_i, \sigma, \tau, (\alpha_\ell, s_\ell, t_\ell)_\ell) = \frac{\Gamma(\beta_0) \prod_{\ell=1}^{K} \alpha_\ell^{\frac{\beta_0}{K}-1} E_\ell}{\Gamma(\frac{\beta_0}{K})^K \alpha^{\beta_0} \prod_{ij} A_{ij}!} \prod_{\ell m} \frac{G(\lambda_a+n_{\ell m}, \lambda_b+T_\ell T_m)}{G(\lambda_a, \lambda_b)} \tag{7}$$

where $G(a, b) = \Gamma(a)b^{-a}$ is the normalization factor of the Gamma distribution and $T_\ell = t_\ell + s_\ell$. Finally notice the $\eta = 1$ case, corresponding to collapsed version of Caron & Fox (2014), can be obtained by taking the limit $\lambda_a = \lambda_b \to \infty$ in which case $\frac{G(\lambda_a+n, \lambda_b+T)}{G(\lambda_a, \lambda_b)} \to e^{-T}$. When discussing the $K = 1$ case we will assume this limit has been taken.

**Inference details**

Sampling the expression eqn. (7) requires three types of sampling updates: For $A_{ij}$ we must apply a sampling procedure to impute missing values, the sequence of block-assignments $(z_i)_i$ must be updated, the parameters associated with the random measure $\sigma, \tau$ must be updated and finally the remaining variables $(\alpha_\ell, s_\ell, t_\ell)$ associated with each expression $E_\ell$ must be updated. We will first consider the later problem:

**Update of variables associated with each $E_\ell$:**   All terms except the densities $g_{\alpha,\sigma,\tau}$ are amenable to standard sampling techniques. In (Caron & Fox, 2014) this expression was sampled by employing a proposal distribution proportional to the density, thus allowing their value to cancel. In our work we opted for the approach of Lomelí et al. (2014) in which $u$ in Zolotarev's integral representation (see main text for details) is considered an auxiliary parameter. Thus, introducing $u_\ell \in ]0, \pi[$ for each $\sigma$-stable random variable gives the full set of variables $\Phi_\ell = (\alpha_\ell, s_\ell, t_\ell, u_\ell)$ for each $E_\ell$ term. For convenience, the domain of the variables are in turn transformed to $\mathbb{R}$ using the standard change-of-variables $x \mapsto e^x$ for $\alpha, t$ and $s$ and the logistic mappings $x \mapsto (1 + e^{-x})^{-1}$, $x \mapsto \pi(1 + e^{-x})^{-1}$ for $\sigma$ and $u$. We found a simple random-walk Metropolis-Hastings sampling with a $\mathcal{N}(0, \sigma = 0.1)$ kernel (50 steps per iteration) was robust and efficient compared to the other updates.

**Update of $z_i$:**   These variables can be updated directly from the likelihood eqn. (7), however we opted to re-impute the weights $(w_i)_{\tilde{z}_i=\ell}$ by inverting the integration step from eqn. (3) to eqn. (4) to obtain

$$(w_i/s_\ell)_{i:\tilde{z}_i=\ell} \sim \text{Dirichlet}\left((n_i - \sigma)_{i:\tilde{z}_i=\ell}\right). \tag{8}$$

Doing this for each $\ell = 1, \ldots, K$ allows all variables $z_i$ to be updated in a regular Gibbs sweep.

**Update of $A_{ij}$:**   Most networks are binary whereas the model assumes count-data. Furthermore to test the model it is useful to predict the presence of unobserved edges. Both of these difficulties are resolved by imputation. Suppose we are given a matrix $W$ such that $W_{ij} = 1$ iff. the edge-count $A_{ij}$ is unobserved. Furthermore assume $A_{ij}$ is binary and must be imputed. Edges can then in principle be imputed directly by performing MCMC updates of $A_{ij}$ and accepting/rejecting according to the likelihood eqn. (7), however the coupling between different counts through the gamma functions in $E_\ell$ would make such a sampling procedure prohibitively expensive. This difficulty is not present in Caron & Fox (2014) where the sociability-vector $(w_i)_i$ are retained and updates using Hamiltonian Monte-Carlo, however we can re-sample $(w_i)_i$ and $(\eta_{\ell m})_{\ell m}$ from their marginal distributions and use the re-sampled values of $(w_i)_i$ to impute the corresponding values of $(A_{ij})$. Thus for each plate $(\ell, m)$ we sample $(w_i)_{\tilde{z}_i=\ell}$ from (8) and $\eta_{\ell m}$ from

$$\eta_{\ell m} \sim \text{Gamma}\big(n_{\ell m} + \lambda_a, (t_\ell + s_\ell)(t_m + s_m) + \lambda_b\big) \tag{9}$$

the distribution of each unobserved $A_{ij}$ is then simply $\text{Poisson}(\eta_\ell w_i w_j)$, $z_i = \ell, z_j = m$.

**Validation of the sampler**

To investigate the validity of the sampling procedure, we considered the $K = 1, \lambda_a = \lambda_b \to \infty$ case and used the sampling procedure of (Caron & Fox, 2014) with $(\alpha = 2, \sigma = 0.5, \tau = 1)$ to generate $250\,000$ random networks. As described in the previous section the probability of any given network is fully determined by the edge-endpoint counts $(n_1, \ldots, n_k)$ and the probability of a particular sequence of counts is permutation invariant. If ordered decreasingly this gives 41 unique vectors of edge-endpoint counts $(n_1, \ldots, n_k)$ for $L = 0, 1, 2, 3, 4$ (see vertical axis on figure 1a) and the generated networks were binned according to their edge-endpoint count signature (networks with more than 4 edges were discarded). In this manner we obtained an estimate of the true frequency of a particular network signature. This estimate of the frequency was compared against the probability of a given network as computed by eqn. (17). Notice that due to permutation invariance the probability of each network signature must be corrected by multiplying eqn. (17) with a factor obtained by a combinatorial argument (see for instance Pitman (2006, eqn. (2.2)))

$$\frac{n!}{\sum_{i=1}^{m_i}(i!)^{m_i}m_i!}, \quad \text{where} \quad m_i = \sum_{i=1}^{k} 1(n_i = 1).$$

(a) Network frequency check      (b) Validation of $f_\sigma$

Figure 1: (Left:) The estimated frequency of all unique networks binned according to their unique edge-endpoint counts $(n_1, \ldots, n_k)$. (ordered decreasingly) for $L = 0, \ldots, 4$ edges (41 in total, red circles), as well as the frequency obtained by computing the probability (see text for details). (Right:) The density of the stick length $\sum_i w_i$ for the randomly generated networks as well as the true density eqn. (11) obtained by numerical integration of eqn. (10)

We thus obtain two estimates of the probability of a particular network signature shown in figure 1a, both in close agreement.

In figure 1b is shown the estimated density of the total mass $T$ obtained by numerically integrating Zolotarev's integral representation of $f_\sigma$

$$f_\sigma(x) = \frac{\sigma x^{\frac{-1}{1-\sigma}}}{\pi(1-\sigma)} \int_0^\pi du \, A(\sigma, u) e^{-A(\sigma,u)/x^{\sigma/(1-\sigma)}},$$

$$A(\sigma, u) = \left[ \frac{\sin((1-\sigma)u)^{1-\sigma} \sin(\sigma u)^\sigma}{\sin(u)} \right]^{\frac{1}{1-\sigma}}, \tag{10}$$

$$g_{\alpha,\sigma,\tau}(t) = \theta^{-\frac{1}{\sigma}} f_\sigma(t\theta^{-\frac{1}{\sigma}}) \phi_\lambda(t\theta^{-\frac{1}{\sigma}}). \tag{11}$$

The estimated density of the total mass $T$ obtained by summing the generated sticks $(w_i)$. Both the estimates of the networks signatures and the density of $T$ are in close agreement.

**Datasets and preparation**

To test the methods we selected 11 publicly available datasets describing social networks, co-authorship networks and biological networks.

**Yeast:** Interaction network of 2361 proteins in yeast (Bu et al., 2003).

**SmaGri:** Coauthorship network of 1059 authors from the Garfield's collection of citation networks (Batagelj & Mrvar, 2014).

**SciMet:** Coauthorship network of 3084 authors from the Scientometrics journal, 1978-2000 (Batagelj & Mrvar, 2014).

**Netscience:** Coauthorship network of 1589 authors working in network theory as compiled by Newman (2006).

**Hagman:** Structural brain networks where edges correspond to the number of fiber tracts between 998 brain regions. All five networks in the dataset were simply averaged to produce a single network (Hagmann et al., 2008).

**NIPS:** Consisting of the 2865 authors who have coauthored papers together at the 1-12'th NIPS conference (Roweis, 2009).

**Caltech, Simmons, Reed, Haverford, Swarthmore:** Five social networks of $769, 1446, 962, 1518, 1659$ students respectively obtained from the Facebook100 dataset (Traud et al., 2011).

The datasets were processes similarly by first removing any vertices without edges, i.e. where $n_i = 0$, and thresholding at 0 to produce binary networks. Selection of the missing edges for link prediction was done by first removing a fraction of $5\%$ of all potential edges at random and then, if this procedure left any vertices without attached edges, re-introducing one of the edges attached to each such vertex and removing (at random from all other potential edges) a single edge. This procedure was repeated until $5\%$ of the potential edges were removed and all vertices had at least one edge attached.

### Models considered

In addition to non-parametric extensions of the Poisson SBM we compared the CRMSBM against a degree corrected block model, the *degree-corrected stochastic block model* (DCSBM) of Herlau et al. (2014). This model is not exchangeable but does model block structure and sociability.

Specifically the DCSBM assumes a generative process of the form:

$$
\begin{aligned}
(z_1, \ldots, z_n) &\sim \mathrm{CRP}(\alpha) \\
\eta_{\ell m} &\sim \mathrm{Gamma}(\lambda_a, \lambda_b) \\
(\theta_{i\ell}^{(1)}), (\theta_{i\ell}^{(2)}) &\sim \mathrm{Dirichlet}((\gamma)_{i=1}^{k_\ell}) \\
A_{ij} &\sim \mathrm{Poisson}(k_{z_i} k_{z_j} \theta_{i z_i}^{(1)} \theta_{j z_j}^{(2)} \eta_{z_i z_j}).
\end{aligned}
$$

To be consistent with the CRMSBM we selected a prior of the form $\mathrm{Gamma}(2,1)$ for $\alpha, \lambda_a$ and $\lambda_b$. The model is somewhat sensitive to the choice of prior for $\gamma$ however we found a prior of the form $\mathrm{Gamma}(2,1)$ to perform reasonably well. The DCSBM reduces to a model without degree-correction, the pIRM (Kemp et al., 2006), by the choice $\gamma_{i\ell} = \frac{1}{n_\ell}$.