[Reviews · NeurIPS 2016]

Reviewer 1

Summary

The paper explores a nice block structure extension to the construction used in Caron and Fox (2014). Ideas from Lomeli et al are also used to propose a collapsed inference scheme.

Qualitative Assessment

A contribution that fits an obvious hole in the literature. The paper is well written except for (many, unfortunately) glaring typos, perhaps introduced during last minute editing? A pleasure to read otherwise. I only have a few minor suggestions: ## Main text - line 33: " will to refer " - "Baysian" -> spellcheck! - it’s density -> its - Palms formula -> Palm's formula - "metropolis hastings" -> "Metropolis Hastings.", missing period too ## Supplement - are retained and updates using Hamiltonian -> updated

Confidence in this Review

2-Confident (read it all; understood it all reasonably well)


Reviewer 2

Summary

This paper extends the construction of Fox & Caron to construct a stochastic model of networks with exchangeable vertex properties (e.g. degree) and that also exhibit latent vertex properties (e.g. clustering). The authors do this by incorporating auxiliary variables through the general representation theorem of Wallenberg. The authors derive the form of the posterior distribution and develop a Gibbs sampler for it. They demonstrate the efficacy of the model on link prediction for several real network data sets.

Qualitative Assessment

Overall, I think this is a nice, well written paper that presents very clean ideas with a very good supplement filling in a lot of the omitted details. I think the experiments for the most part demonstrate the the model performs comparably to other state of the art models and in some cases does better. This being said, the authors themselves say that the extension is relatively straight-forward, and it's unclear how much the proposed framework adds to the practical modeling of networks beyond the fact that the model is exchangeable (the authors may want to focus on why this is an important invariant for network modeling). I have a couple comments that I think could help the paper: - There are a lot of spelling and grammatical errors. - In Fig. 2, the axis for Steps 1 and 3 are unreadable. I think the figures could be extremely useful to readers, but without labels I had to guess what they were trying to depict. - In Fox & Caron, sampling the \sigma parameter of the GGP turned out to be rather difficult, was this the case for the proposed algorithm? Or did the fact that the authors could collapse out particular parameters make that parameter easier to sample? - In Fig. 4b, why is it a line plot? The x-axis is categorical, connecting the lines makes no sense. Make it a bar plot or something appropriate. - The authors state in the caption of Fig. 4 that the lines are the averages of 4 restarts. This should be put in the main text when the authors mention they do 4 restarts as it's important to reproducing the results. Also, why is there no uncertainty displayed on the figure? If it's very small this should be stated? -

Confidence in this Review

2-Confident (read it all; understood it all reasonably well)


Reviewer 3

Summary

The paper is concerned with an extension of Caron & Fox (2014) idea of using completely random measures to model sparse graphs. The extension includes block structure which cannot be directly modeled in Caron & Fox (2014) setting. This is of obvious interest since most of real data networks display heterogeneity, for instance in term of degree distribution, which can be handled by adding block structure. To the aim of introducing block-structure, each location of the CRM is extended to include a uniform variate, which work as allocation variables given a block partition of the space.

Qualitative Assessment

The clarity of the paper could be much improved. Typos are very numerous. Most of them were already reported in an earlier review but were kept unchanged in this version (http://robjhyndman.com/hyndsight/always-listen-to-reviewers/). Comments: - the way block-structure is introduced was not immediately clear to me. This could be made clearer from the end of Sec 2.1 for example. - the choice of the number of blocks K is not discussed. How should it be fixed in the experiments? - what is k in bottom row of Figure 3? Typos: - accordance of singular-plural between subject and verb is very often incorrect (see eg 3rd paragraph of Sec 4, but also throughout the text) - inconsistent use of brackets for citations in text  - in Fig 4 (a), only 4 curves are visible out of 6 - References are poorly proofread:  - inconsistent use of first names (with or without initials) - inaccurate references (eg Lomeli et al. is published) - in appendix, remove "et al." in "Lijoi, Antonio, Prünster, Igor, Walker, Stephen G, et al."

Confidence in this Review

3-Expert (read the paper in detail, know the area, quite certain of my opinion)


Reviewer 4

Summary

Past research had introduced the concept of exchangeability in network models, whereby two isomorphic graphs have the same distribution. In particular, Caron & Fox (2014) incorporated power-law degree inhomogeneity to exchangeability of graph models. The present paper proposes a technique to model exchangeability and edge-inhomogeneity (power-law degree distribution) but, in this case, the model incorporates also block-structure in networks.

Qualitative Assessment

While the idea of modeling exchangeability for block-structure is not new in itself (see for instance Nowicki and Snijders 2001 for statistical perspective of the same problem), the perspective the authors follow seems novel because they extend Kallenberg exchangeability to model block-structure. While this approach can help elucidate exchangeability in network models I have a few concerns about the empirical evidence shown in the paper and the presentation of sections 2.2-2.5. Some drawbacks: Technical: The experiments state that the choice of parameters for the competing models is "similar" to the choice for the proposed model without providing the experimental setup for the alternative models. However, this should be explicitly explained for replicability purposes. The results themselves seem not very convincing. For instance, in figure 4b the performance of the proposed model DCSBM seem not very much better than that of competing alternatives, except for the first 3 datasets. However, and more importantly, not much can be said of the advantage of DCSBM because the results are for a single run (with thousands of iterations of the algorithm). Hence, without an statistical assessment of the results it is hard to tell what is the advantage of the proposed model with respect to the other alternatives (for instance Nowicki and Snijders (2001)). Lastly, the conclusion could be improved by explicitly stating how much reduction of the number of parameters (line 284) is attained with the proposed model. Presentation: In general Sections 2.2-2.5 specialy 2.4 and 2.5 are hard to follow. I would kindly recommend to review these sections. The reason for this is that some notational information is not explicitly explained, or has been omitted. For instance: All symbols used in the paper must be explicitly defined including $\delta$ (Dirac measure?). This also applies to operations, e.g. $o$ or $\otimes$ in line 113 or $dw$ and $d\theta$. Please, kindly explicitly state what basic properties of completely random measures (CRM) make the thinned measures in line 172 also CRM (Don't leave this to the reader). The authors do not define what the equality in line 40 means (equality in distribution?, I assume). The paper leaves many details (e.g paragraph 190-195) to outside sources which makes it not fully self-contained. Some minor typos: The sentence 32-33 "will to refer to". Many openning/closing brackets are of the type [a,b [ (this may be a type unless the authors meant open intervals, which is not stated in the paper): e.g. line 126, line 180. Possible should replace possibly in line 110, and vertex should replace vertice in line 93

Confidence in this Review

2-Confident (read it all; understood it all reasonably well)


Reviewer 5

Summary

The authors present a block model extension of the CRM sparse graph by Caron + Fox model based on Kallenberg exchangeability of random measures. They develop a alternative inference algorithm based on counting arguments and evaluate their sampler’s autocorrelation and compare their model on link prediction to the pIRM, degree corrected SBM, and the Caron and Fox model. They generally perform best.

Qualitative Assessment

In general, I found the paper well written and easy to follow. The extension the authors propose is a natural one and give the long history of block modeling one worth pursing. One concern I have with the paper is that the evaluation favors exchangeable models in the kind of subsampling applied to a graph tells you things about the invariances the subsampled graph will have. Is there another evaluation that can be added without this? Next it feels like equation 4 is the product of a graphon a factorized graphex (to borrow from Veitch and Roy). Is there an advantage to this general form over the graphex or simply the discrete case which forms the motivating example? Finally, I’d like to see visualizations of the community structure found with this model. How similar is it to the exchangeable or degree corrected block models? This is important as visualizations form the basis for how users in social science and biology use these kind of models. Minor: 1) Line 76-81 looks like its for when xi = Lebesgue. That should be be mentioned 2) Line 293 dependent and correlated random measures don’t mean exactly the same thing. It’s good to be precise.

Confidence in this Review

2-Confident (read it all; understood it all reasonably well)


Reviewer 6

Summary

The authors consider the problem of modeling real-world data showing both power-law degree distributions and a block-structure. They first consider the model of (Caron&Fox,2014), which uses completely random measures and a point process representation of networks to build models with power-law, and they extend it by incorporating a block-structure. Then they derive an expression for the posterior distribution where some variables are integrated out analytically, to obtain more efficient Monte Carlo methods. Finally they compared the performances of their model in terms of missing edge predictions with the original Caron&Fox model and some other block-structure models on 11 datasets. The results suggests that block-structured models perform much better than the original Caron&Fox model and that the proposed model performs marginally better than the other block-structured models considered.

Qualitative Assessment

I enjoyed reading the paper. The generalization of Caron&Fox model, despite being simple, seems to be justified and appropriate. This is supported by the fact that the prediction results obtained are positive. I have only minor remarks. -From the exposition point of view, the number of typos could be slightly reduced (e.g. line 59) and Figure 4 should be made more readable when printed in gray scale. -Why does \mu has no indices and still an iid after line 176? -If possible, it would be nice to have a short justification of the priors chosen (e.g. do the Gamma(2,1) priors induce reasonable prior expectation and variances on quantities of interest?). Are the results sensitive to the choice of priors? - Equation (5): I think the general characterization of CRM includes also deterministic parts that can be continuous, so the statement at lines 152-154 is not correct I believe. It would be enough to include "non-trivial" or "non-deterministic". - Figure 4 should be made more readable when printed in gray scale if possible - Do you state anywhere how long the MCMC took to run on the datasets? It would be a useful information to have. It would also be nice (although not strictly necessary) to compare such runtime with the one of Caron&Fox as you claim that integrate w_i produces significant improvements.

Confidence in this Review

1-Less confident (might not have understood significant parts)